

# Taking advantage of the software product line paradigm to generate customized user interfaces for decision-making processes: a case study on university employability

Andrea Vázquez-Ingelmo[1], Francisco J. García-Peñalvo[1] and Roberto Therón[1,2]

[1] GRIAL Research Group, Department of Computer Science and Automatics, University of Salamanca, Salamanca, Spain

[2] VisUSAL Research Group, Department of Computer Science and Automatics, University of Salamanca, Salamanca, Spain

Corresponding author
Andrea Vázquez-Ingelmo,
andreavazquez@usal.es

## ABSTRACT

University employment and, specifically, employability has gained relevance since research in these fields can lead to improvement in the quality of life of individual citizens. However, empirical research is still insufficient to make significant decisions, and relying on powerful tools to explore data and reach insights on these fields is paramount. Information dashboards play a key role in analyzing and visually exploring data about a specific topic or domain, but end users can present several necessities that differ from each other, regarding the displayed information itself, design features and even functionalities. By applying a domain engineering approach (within the software product line paradigm), it is possible to produce customized dashboards to fit into particular requirements, by the identification of commonalities and singularities of every product that could be part of the product line. Software product lines increase productivity, maintainability and traceability regarding the evolution of the requirements, among other benefits. To validate this approach, a case study of its application in the context of the Spanish Observatory for University Employability and Employment system has been developed, where users (Spanish universities and administrators) can control their own dashboards to reach insights about the employability of their graduates. These dashboards have been automatically generated through a domain specific language, which provides the syntax to specify the requirements of each user. The domain language fuels a template-based code generator, allowing the generation of the dashboards' source code. Applying domain engineering to the dashboards' domain improves the development and maintainability of these complex software products given the variety of requirements that users might have regarding their graphical interfaces.

## INTRODUCTION

The concept of employability has increasingly gained relevance over the last decades. There is a reason: knowing which factors increase the possibility to obtain a job or to perform better in current job positions could be decisive to improve individual and collective life quality.

However, this concept is still far away from having a straightforward definition (*Chadha & Toner, 2017*). As the literature suggests, employability can be seen as a capability to gain employment or as a set of skills and knowledge required to perform effectively in the workplace, among other definitions (*Universities UK & Confederation of British Industry, 2009*; *Hillage & Pollard, 1998*; *Yorke, 2006*). This lack of consensus when defining employability makes the research in this field a complicated task, given the fact that the definition of its factors depends on the perspective used to evaluate it, as well as the socioeconomic context in which employability and employment studies are framed. For these reasons, nowadays research on employability asks for an exploratory approach, to build stronger theoretical foundations.

Researching on employability has many potential benefits, aiming not only at knowing the variables that affect the capability to gain employment and have a successful work career, but also to exploit this knowledge to help policymakers and institutions with their missions. This knowledge can contribute to the creation of greater policies, focusing on the detected factors to enhance people's chances to obtain better employment. Specifically, educational institutions like universities could benefit from this knowledge. These institutions play a vital role regarding the employability of individuals (*García-Peñalvo, 2016*), as they are in charge of transmitting knowledge and a series of skills to their students. By promoting the most relevant skills and capabilities that affect employability, it could be possible to increase the alignment of education with the labor market.

However, generating knowledge in such a study field is not a trivial task. As it has been introduced, there could be several variables involved in the research of students' employment and employability, so it is necessary to collect significant data volumes to be able to reach valuable insights. In addition to data collection, performing data analysis (*Albright, Winston & Zappe, 2010*) is required to be able to reach useful insights. It is worth noting that analyzing employability data to identify and understand its factors could become a cornerstone in decision-making processes within educational institutions.

Nevertheless, even after performing data analysis, identifying patterns and indicators derived from the analysis outcomes remains a complex challenge. That is why it is crucial to assist decision-makers with powerful tools that allow reaching insights about the domain of the problem, to support decisions with complete and quality information (especially in the academic context, where these processes might have a series of social implications), that is, information and knowledge that has been extracted through visual analysis.

Information dashboards are one of the most commonly used software products for visual data analysis and knowledge extraction (*Few, 2006*; *Sarikaya et al., 2018*). In a domain like employability, these tools can support exploratory studies through a set of graphical and interactive resources, allowing users to envision data more understandably (*Tufte & Graves-Morris, 2014*) and identify relevant relations, indicators or patterns among

large sets of data. It is essential to bear in mind that information dashboards are not just a set of aesthetic graphs and visualizations; they have to effectively transmit information to answer the questions of the users regarding the target domain. Moreover, this is not a trivial job, because of two main reasons: data and users themselves.

On the one hand, users do not have a set of standard and static requirements; they could demand different features or design attributes given their specific goals or needs. On the other hand, data is continuously increasing and evolving nowadays, so it is foreseeable that new information requirements will arise in time. Returning to the employability subject, information requirements in this domain might change in many different ways as this concept could demand new kind of variables or larger amounts of data to explore emerging dimensions or to perform more in-depth analyses.

For these reasons, information dashboards not only need to be useful concerning functionality but also be customizable to adapt to specific user requirements. Also, they should be flexible and scalable regarding its data sources and structures, making the development and maintenance of information dashboards even more complicated. Of course, these issues could be addressed by developing particular dashboards for each involved user to achieve every specific goal, but clearly, this solution would be time-consuming and would require a lot of resources during the development and maintenance phases. Also, scalability would be almost impossible, as new users or changes in the requirements would necessarily imply more resources.

There are, nevertheless, a series of strategies to deal with these challenges. Specifically, software engineering paradigms like software product lines (*Clements & Northrop, 2002*; *Gomaa, 2004*; *Pohl, Böckle & Van der Linden, 2005*) provide powerful theoretical frameworks to address flexibility, scalability and customization in software products that share sets of features within a common domain. Through the analysis of commonalities and variability points in the product domain, it would be possible to reduce the development and maintenance effort of building tailor-made solutions. This paradigm is potentially applicable to dashboards since these software products could be factored into sets of configurable components with configurable features. This paper describes the application of the SPL methodology to the dashboards' domain through the study of their characteristics and the definition of a DSL to manage the product derivation automatically. The main focus of this research is to test the potential usefulness and feasibility of this approach to manage fine-grained features that can be scattered through different code assets, and consequently, to provide a base method for generating personalized dashboard solutions to fit concrete user requirements.

The remainder of this work is structured as follows. Background discusses the background of the problem of generating customized dashboards as well as their application to the employment and employability domain. Context presents the application context and the motivation behind this pilot framework to generate dashboards to support visual analysis on university employment and employability data (framed within the Spanish Observatory for University Employability and Employment studies. Materials and Methods describes the techniques used for the development of an initial approach to a generative dashboard framework. Finally, the Results section exhibits the outcomes of this research to

conclude with the discussion of the developed SPL and the conclusions derived from these results.

## BACKGROUND

The main idea behind software product lines (SPLs) is that the final products can be derived from a set of configurable core assets, allowing their adaptation to fit specific requirements. These core assets are developed during the domain engineering phase, in which commonality and variability of the target product domain are identified to build a common base of components. Core assets are developed with variability points in which specific functionalities could be injected to obtain new products. Functionalities in SPLs are seen as features; the combination of the defined features within the scope of the line (generally following a feature model (*Kang et al., 1990*) allow stakeholders to build personalized products by reusing and assembling software components.

The SPL paradigm has been applied to a variety of domains: Mobile applications (*Marinho et al., 2010*; *Nascimento, 2008*; *Quinton et al., 2011*); Applications for visualizing population statistics (*Freeman, Batory & Lavender, 2008*); Sensor data visualizations (*Logre et al., 2014*); Variable content documents (*Gómez et al., 2014*); or e-Learning systems (*Ezzat Labib Awad, 2017*).

These practical applications have proved the benefits of this paradigm. However, features usually refer to the software's logic, deflecting attention to the presentation layer. The idea of generating customized dashboards can be seen as a specific case of graphical user interfaces (GUI) automatic generation within SPLs. User interfaces require additional work regarding their implementation; they not only need to be functional but also usable to allow users to complete their tasks efficiently and achieve their goals. That is why the design of user interfaces is present through the whole development process, being time- and resource-consuming job.

Automation regarding GUI generation in software product lines has already been faced in several works. Generally, there is a lack of usability on the generated products that can be addressed by manually designing every product GUI. But this approach is highly inefficient in the SPL paradigm context since all the development time saved could be lost by introducing a manual task (*Hauptmann et al., 2010*). Integration of the GUI design process and the SPL paradigm is required to leverage the benefits of the two approaches (*Pleuss, Botterweck & Dhungana, 2010*). There is, as *Pleuss et al. (2012a)*; *Pleuss et al. (2012b)* pointed out, a dilemma between automation and usability. To address this challenge, they utilized Model-Based UI Development (MBUID) methods to separate the functionality and the appearance of the GUI (*Pleuss, Botterweck & Dhungana, 2010*).

On the other hand, *Gabillon, Biri & Otjacques (2015)* demonstrated the possibility of creating adaptive user interfaces through the Dynamic SPL (DSPL) paradigm and MBUID models by developing a context-aware data visualization tool that can be adapted during runtime.

DSPLs provide a useful paradigm for adapting code at run-time, obtaining adaptive GUIs. *Kramer et al. (2013)* proposed document-oriented GUIs with run-time variations

through XML documents (*Kramer et al., 2013*). This context-adaptable feature has also been achieved by *Sboui, Ayed & Alimi (2018)*, by developing a mobile application that is also runtime adaptable through MBUID models and reusable artifacts. In this particular case, the code generation is based on eXtensible Stylesheet Language Transformations (XSLT) and XML files (*Sboui, Ayed & Alimi, 2018*). These works shows not only the viability of GUI generation in the SPL/DSPL paradigms context but also their valuable benefits.

It seems evident that GUI customization requires fine-grained features to achieve the desired usability and design attributes. Fine-grained features mostly require annotative approaches regarding their implementation, given their specialization. Annotative approaches can address this issue because annotations can be arbitrarily specified at different source code fragments (*Kästner & Apel, 2008*; *Kästner, Apel & Kuhlemann, 2008*), and provide a framework for fine-grained automated software composition through feature structure trees (*Apel, Kastner & Lengauer, 2009*).

There are different approaches to manage the implementation of variability at a fine-grained level (*Gacek & Anastasopoules, 2001*). Especially, frame- and template-based approaches provide valuable solutions to address this fine-grained level of variability, allowing the injection of particular fragments of code at any point of the base source code. Frame-based languages, like XML-based Variant Configuration Language (XVCL) (*Jarzabek et al., 2003*), provide a syntax to combine and insert fragments of code through the definition of frames, allowing the separation of concerns regarding the SPL implementation (*Zhang, Jarzabek & Swe, 2001*). Templating can also achieve valuable results; templating libraries such as Jinja2 (*Ronacher, 2008*) provide powerful functionalities to annotate the source code independently of the target programming language (*Clark, 2018*; *Ridge, Gaspar & Ude, 2017*).

The generation of GUI within the context of a product family is still a convoluted field, although the previous work has enlightened the path to improve and leverage the automation and generation of these complex software elements. The complexity mainly comes from human factors and the vast variety of requirements regarding user interfaces.

This work aims to present an application of the SPL paradigm, in this case on the dashboards' domain, considering the fine-grained nature of their features and the necessity of customizing its interaction methods and visual appearance.

## CONTEXT

The application of this work is framed within The Spanish Observatory for University Employment and Employability. The following subsections describe this organization's mission and the motivation to generate personalized dashboards to explore its data.

### The observatory for university employment and employability

The Observatory for University Employment and Employability (also known as OEEU, its Spanish acronym, http://oeeu.org) is an organization with the vision of becoming an information reference for understanding and exploiting knowledge about employment and employability of students from Spanish universities. To do so, this network of researchers and technicians conduct studies about these fields in the academic context (*Michavila et*

*al., 2018a*; *Michavila et al., 2016*; *Michavila et al., 2018b*), through a data-driven approach to recollect, analyze, visualize and disseminate employment and employability data of graduates from Spanish universities.

Firstly, in the data collection phase, universities provide their administrative records and, once this phase is completed, their students answer a questionnaire about different aspects of their education and work career. This process leaves the Observatory with a significant set of variables from the students' sample. For instance, in the 2015 study edition, more than 500 variables were gathered from 13,006 bachelor students. Moreover, in the 2017 study edition, 376 variables were gathered from 6,738 master degree students.

The volume of the data collected makes the presentation of the study results to the Observatory ecosystem's users a challenge, as the latter may have different requirements and necessities regarding the studies' data. For these reasons, an approach based on domain engineering fits the OEEU's needs, allowing an efficient generation of customized dashboards that meet different requirements.

## Motivation

As it has been introduced, employment and employability are complex study fields that mainly ask for exploratory analysis, given its relatively initial status of research. In the context of the Spanish Observatory for University Employment and Employability, where a vast set of variables from significant quantities of students are recollected, it is crucial to rely on exploratory visualizations that allow users and administrators to identify at a glance unusual patterns or important data points by enhancing the understanding of the presented information (*Card, 1999*).

In contrast with explanatory visualizations, in which the primary purpose is to tell a story through data, exploratory tools aim to facilitate users to pose more questions as data is being explored. In essence, explanatory analyses start from a question and use data to answer it. Exploratory analysis, on the other hand, uses data to detect new avenues of research. For instance, when a user does not have a clear question about the data, it will use exploratory research to find patterns or relations among variables. This same user could employ the acquired knowledge to explain the insights reached through previous explorations using an explanatory visualization.

Exploratory visualizations rely intensely on interaction to provide their functionality and to allow users to drill-down datasets, being able to discover new aspects of the domain by directly communicating with the graphical interface. However, an interaction can take many forms, and there is not a single solution to obtain usable and intuitive interfaces valid for every user.

For instance, some users could find useful a visible control panel to manage data if they are going to apply filters, aggregations and so on intensively. On the other hand, other users can demand in-place interaction if they give more importance to having more space for the visualizations (instead of having a permanent control panel consuming screen space). Another example is that users that speak a left-to-right (LTR) or a right-to-left (RTL) language would demand different layouts for the same task, according to their sociodemographic or cultural context (*Almakky, Sahandi & Taylor, 2015*; *Marcus & Gould,*

*2000*). Also, visualization novices could require task-oriented dashboards to support their visual analysis, since their past experience with this kind of tools is a relevant factor when interacting with a system (*Elias & Bezerianos, 2011*).

Once patterns, relations between variables and interesting dimensions have been identified through the exploration of data, even the exploratory nature of a dashboard can change for a more explanatory purpose to present the results understandably and strikingly.

For all these reasons dashboards, their components, their interaction, and even their primary purpose need advanced configuration and customization to fit into different contexts and requirements. Moreover, as it has been aforementioned, SPLs provide a potential solution to efficiently address this customization since visual components and interaction methods could be treated as features of the product line, decreasing the resources needed during the development and maintenance of dashboards.

# MATERIALS & METHODS

This section presents the materials and techniques used during the development of this first approach to a framework for generating dashboards to explore employment- and employability-related variables.

## Meta-model

The problem to address requires abstract modelling to capture basic features within the dashboards' domain. To do so, a meta-model is proposed. Meta-models are a crucial artefact in model-driven engineering and model-driven architectures (*Kleppe, Warmer & Bast, 2003*), as they allow to define a high-level view of the domain without depending on specific technologies. Therefore, meta-models should remain as simple as possible to eventually, through a series of mappings and transformations, obtain concrete models (*Álvarez, Evans & Sammut, 2001*).

For this generic dashboards' domain, the meta-model found in Fig. 1 is proposed. First of all, a specific user could handle a dashboard. This dashboard could be composed of one or more pages, being these last composed, in turn, by one or more containers. A container could be seen as a row or a column, and it can recursively contain more containers. The container recursion ends with a component, which is any graphic element that can be used in a dashboard. The recursion mentioned above allows the arrangement of any layout by the recurrent combination of rows and columns.

This meta-model eases the vision of the dashboards' domain, and it also allows to identify the common base of any dashboard.

## Feature model

The meta-model gives a high-level vision of the dashboards' domain. However, it does not capture concrete features. That is why software product lines rely on feature models (*Kang et al., 1990*) to identify common and variable assets.

Feature models not only serve as a documentation element but also as an important artifact within the development process. The implementation of the core assets and the materialization of variability points on the code must be guided by the previously defined feature model.

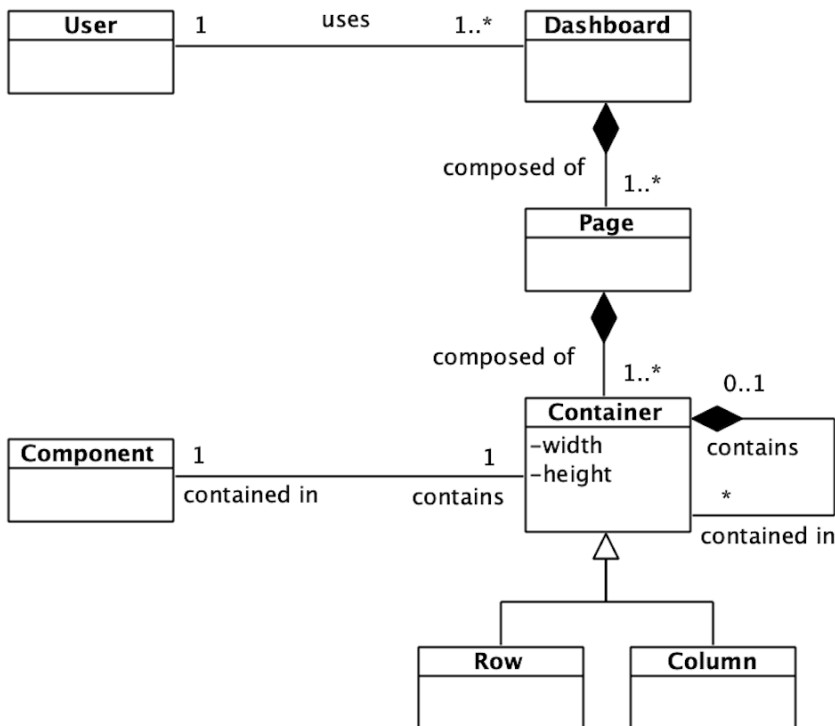

**Figure 1** **Dashboard meta-model.** The dashboard meta-model allows a high level view of the target do-main.

In this domain, the feature model will capture the dashboards' visualization components, as well as individual features and restrictions of each visualization. The hierarchical structure of the feature model allows to define high-level characteristics and refine them through the tree structure until reaching the lower-level features (i.e., fine-grained features). This structure makes the scalability of features easier, since adding new features involves the addition of new nodes to the feature tree uniquely.

For the Observatory's dashboards, three main configurable visual components (features) have been defined: a scatter diagram, a chord diagram and a heat map. These visualizations address the requirements of the Observatory's data but can be reused for other data domains. Also, it is possible to specify a global filter that affects the data of all components previously defined. These high-level features of the dashboards' product line are presented in Fig. 2.

A detailed view of the scatter diagram feature can be seen in Fig. 3. It has a set of subsequent features, either mandatory, optional or alternative. One mandatory feature is the base logic of the scatter diagram (i.e., the component layout construction and its primary logic). Another mandatory feature is the initial data that the diagram will be showing on different dimensions since it must be specified. Among the optional features, it is possible to determine whether a tooltip will show up when hovering on data points if a set of controls will support the data exploration, or the capacity to zoom or export the diagram. Also, a title for the visualization can be included.

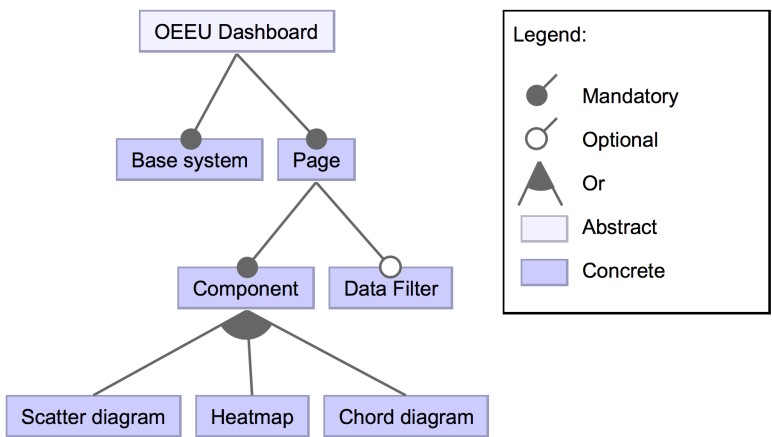

**Figure 2** **High-level view of the feature diagram.** This feature diagram shows high-level components that could compose the dashboard.

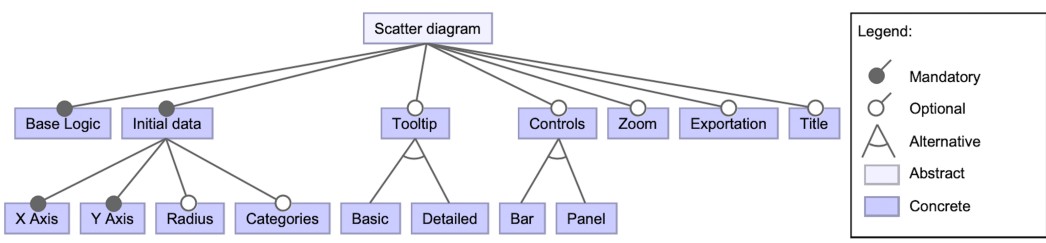

**Figure 3** **High-level view of the scatter diagram component's features.** This snippet of the feature model shows the possible features regarding the scatter diagram component.

For the sake of simplicity, some of the lower-level features have been omitted in Fig. 3. For instance, the bar and panel control features have subsequent features. The detailed features for a panel type control are shown in Fig. 4 to provide an example. A control panel will rely on its underlying logic, and it can count on different optional features, like data selectors to dynamically change the visualization's presented data; in case of the X and Y axes, these selectors could be located within the control panel space or in-place controls (i.e., situated near the scatter diagram axes). Other possible features involve having an overview that shows a detailed view of a data point when hovering, data filters, among others.

The feature diagram provides a high-level and organized overview of the SPL, improving the organization of the source code and development tasks.

## Domain-specific language

There is, however, a necessity of connecting the previous models to the dashboards' source code to be generated (*Voelter & Visser, 2011*). A Domain-Specific Language (DSL) has been designed to accomplish this connection. This DSL is based on the identified domain's features, by structuring them with XML technology (*Bray et al., 1997*) and by validating the model restrictions with an XML schema (*Fallside, 2000*). XML technology provides

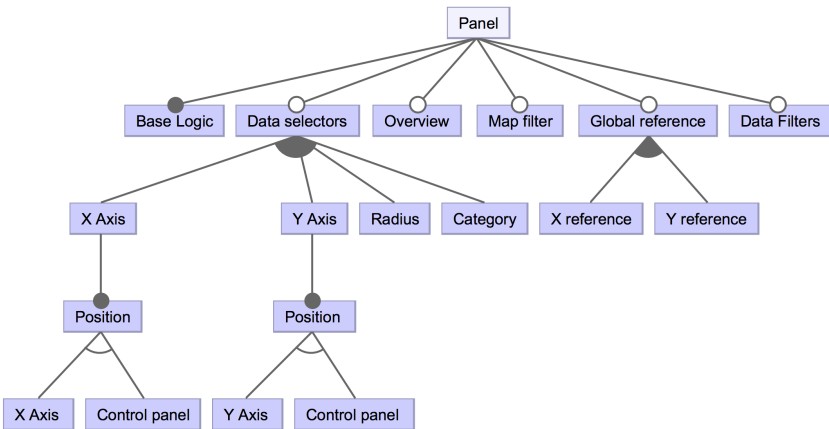

**Figure 4 High-level view of a component's panel subsequent features.** This part of the feature diagram shows lower-level features regarding the components' control panel.

```xml
<xs:element name="ScreensConfig">
    <xs:complexType>
        <xs:choice>
            <xs:element name="PageGroup"...>
            <xs:element name="Page">
                <xs:complexType>
                    <xs:sequence>
                        <xs:element name="DataFilter" minOccurs="0">
                            <xs:complexType...>
                        </xs:element>
                        <xs:element name="Components"...>
                        <xs:element name="Layout" type="LayoutType"/>
                    </xs:sequence>
                    <xs:attribute name="page_id" type="xs:string"/>
                </xs:complexType>
                <xs:unique name="unique-page_id-2">
                    <xs:selector xpath="Page"/>
                    <xs:field xpath="@page_id"/>
                </xs:unique>
            </xs:element>
        </xs:choice>
    </xs:complexType>
</xs:element>
```

**Figure 5 Snippet of the DSL schema.** It is possible to specify the dashboard layout and its elements (i.e., data filters, components, etc.).

a readable and easy-to-parse manner for injecting functionalities or requirements in a system, fostering flexibility since these rules are not directly defined (or hard-coded) in the source code.

The following examples describe the DSL developed for this work. Following the meta-model, every dashboard will be composed by one or more pages, each page with its configuration (i.e., layout and components, as seen in Fig. 5), and each page component with its setting (given the feature model, as seen in Fig. 6).

Data resources of each visual component are represented by the XSD generic type "*anyType*", to decouple the data structure and format from the presentation, and also to open up the possibility of injecting dynamic data sources without affecting the DSL syntax.

```
<xs:element name="Components">
    <xs:complexType>
        <xs:sequence>
            <xs:element name="Component" maxOccurs="unbounded">
                <xs:complexType>
                    <xs:choice>
                        <xs:element name="ScatterDiagram"...>
                        <xs:element name="Heatmap"...>
                        <xs:element name="ChordDiagram"...>
                    </xs:choice>
                </xs:complexType>
            </xs:element>
        </xs:sequence>
    </xs:complexType>
</xs:element>
<xs:element name="Layout" type="LayoutType"/>
```

**Figure 6** **DSL schema regarding the specification of the dashboard components.** It is possible to see the link between the feature model elements and the XML schema elements (e.g., the components that could compose the dashboard).

```
<xs:element name="ScatterDiagram">
    <xs:complexType>
        <xs:sequence>
            <xs:element name="Title" type="xs:string"...>
            <xs:element name="Zoom" type="xs:string"...>
            <xs:element name="Exportation"...>
            <xs:element name="InitialData"...>
            <xs:element name="Controls" minOccurs="0"...>
            <xs:element name="Tooltip" minOccurs="0"...>
        </xs:sequence>
        <xs:attribute name="component_id"
                      type="xs:string"/>
    </xs:complexType>
</xs:element>
```

**Figure 7** **DSL schema regarding the specification of the scatter diagram component.** This part of the DSL represents the available features for the scatter diagram component.

In Figs. 6 and 7 the resemblance of the XML schema structure with the feature model can be appreciated. The hierarchical nature of XML matches with the hierarchical structure of feature diagrams. This resemblance allows better traceability of the features involved in the product line, because the syntax of the DSL is obtained from the feature model, thus providing a computer-understandable specification of the SPL, necessary to process the requirements and to automate the dashboard generation. In this current approach, the dashboard's feature model serves as documentation, but, as it will be discussed, it would be extremely valuable to create a programmatic link between this model and the DSL specification, in order to propagate and reflecting any feature model change automatically in the DSL, improving maintainability.

Finally, Fig. 8 shows how the layout of the dashboard is specified in terms of rows, columns and components (following, again, the meta-model previously presented). The

```xml
<xs:complexType name="LayoutType">
    <xs:choice>
        <xs:element name="RowGroup" type="LayoutType"/>
        <xs:element name="ColumnGroup" type="LayoutType"/>
        <xs:element name="Row" type="LayoutType" maxOccurs="unbounded"/>
        <xs:element name="Column" type="LayoutType" maxOccurs="unbounded"/>
        <xs:element name="Component">
            <xs:complexType>
                <xs:simpleContent>
                    <xs:extension base="xs:anySimpleType">
                        <xs:attribute name="ref" type="xs:string"/>
                    </xs:extension>
                </xs:simpleContent>
            </xs:complexType>
        </xs:element>
        <xs:element name="DataFilter">
            <xs:complexType>
                <xs:simpleContent>
                    <xs:extension base="xs:anySimpleType">
                        <xs:attribute name="ref" type="xs:string"/>
                    </xs:extension>
                </xs:simpleContent>
            </xs:complexType>
        </xs:element>
    </xs:choice>
    <xs:attribute name="width" type="xs:string" use="optional"/>
    <xs:attribute name="height" type="xs:string" use="optional"/>
</xs:complexType>
```

**Figure 8** **XML type for specifying the dashboard's layout.** The dashboard layout (previously modeled through the dashboard meta-model) is specified by creating a custom type.

DSL combines both the meta-model and feature model designs to obtain a specific syntax to configure all the aspects regarding the generation of final products.

The whole schema for the DSL can be consulted at the following GitHub repository https://github.com/AndVazquez/dashboard-spl-assets (*Vázquez-Ingelmo, 2018*).

## Code generator

To put together all the developed assets and concepts, a code generator has been developed to manage the generation of functional dashboards. The generator interprets the DSL (i.e., XML configuration files) and selects the appropriate template (i.e., core assets of the SPL) to configure them by injecting the chosen features, obtaining the dashboards' final source code. The code templates and XML configuration files are managed by the developers following the elicited user requirements.

The inputs and outputs of the code generator can be seen in Fig. 9.

## Code templates

The next challenge regarding the implementation of this SPL involves the choice of the techniques for materializing the product line's variability points. In this case, personalization is focused on the visual elements of the system's presentation layer, which require fine-grained variability (*Kästner & Apel, 2008*). Coarse-grained variability involves the addition and removal of full components, which is also useful for this approach (users may prefer a scatter diagram over a chord diagram to achieve their goals, removing the last from the dashboard). However, visual components themselves (referring to the elements that compose them) also require high variability to fit into different requirements,

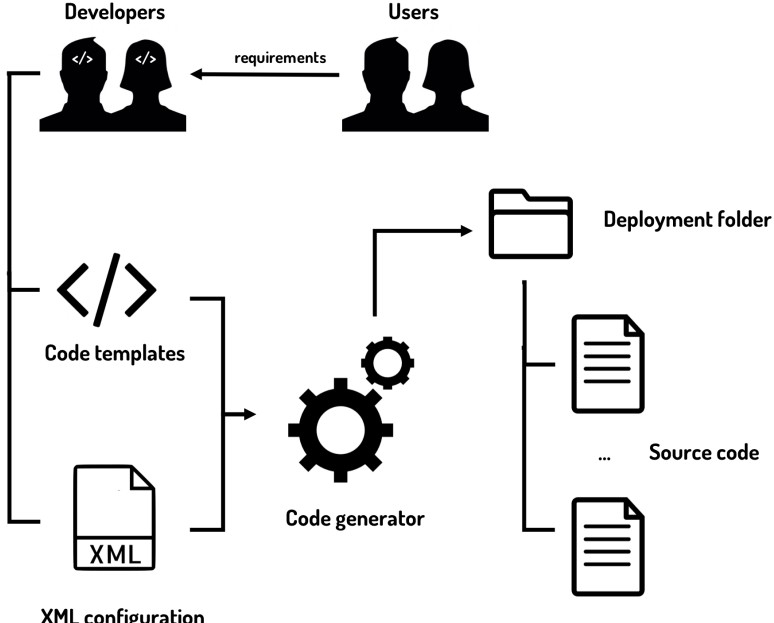

**Figure 9 Code generator inputs and outputs.** The code generator is fed with the code templates and the XML configuration files to provide the final source code of the dashboard.

so fine-grained variability needs to be accomplished. There exist different approaches to implement fine-grained software composition, as in the case of FeatureHouse (*Apel, Kastner & Lengauer, 2009*), which uses superimposition and feature structure trees (FSTs), however, not every method supports the currently required granularity, which involves even statement-level variability. Fine granularity often prohibits superimposition approaches (*Apel, Kästner & Lengauer, 2013*).

The mechanism chosen to reach the desired feature granularity is based on template engines. Template engines allow to tag sections and parameterize units of source code to inject concrete values later and obtain complete source files. This mechanism accomplishes the necessity of materializing the variable features of a tangible product of the line.

Jinja2 (*Ronacher, 2008*) was selected as the engine for developing the core assets of this SPL. This template engine allows the definition of custom tags, filters and even macros, being the last one of the essential features to organize the core assets. As described in (*Kästner & Apel, 2008*), fine-grained approaches can make the source code tedious to read and maintain. By declaring every variant feature on different macros to compose them subsequently, it is possible to achieve high cohesion and loose coupling on the SPL feature implementation process, improving reusability and source code organization by grouping the different functionalities by its parent feature. There was no need to implement extensions of the Jinja2 implementation and mechanisms, as its current syntax was sufficient for the annotative approach followed.

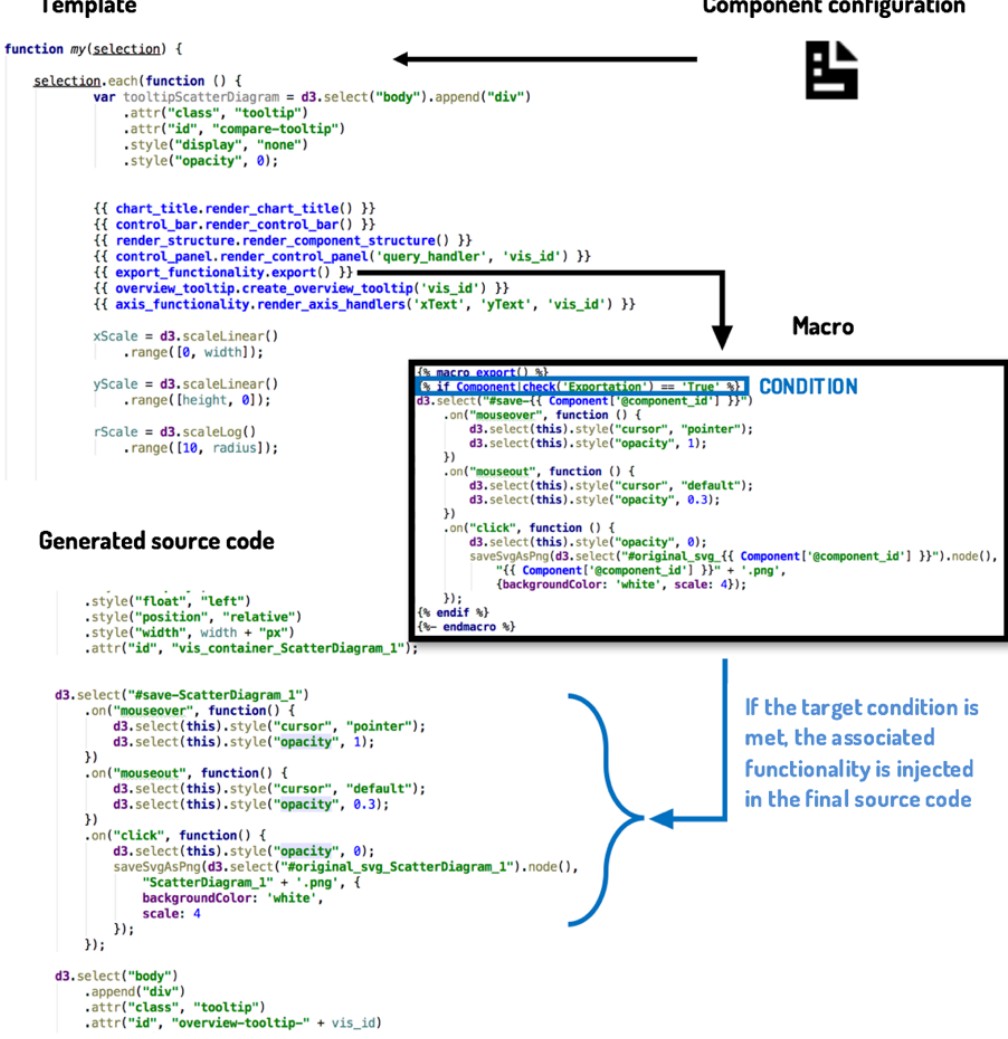

**Figure 10** **Workflow of the code generation process.** A simplified view of the code generator behavior.

A diagram of the detailed workflow for generating the source code can be seen in Fig. 10. The code templates for this case study can be also consulted at https://github.com/AndVazquez/dashboard-spl-assets (*Vázquez-Ingelmo, 2018*).

# RESULTS

## Generated dashboards

As it has been already introduced, the Observatory collects important datasets to research the employability and employment of graduates from Spanish universities. Relying on a customizable exploratory tool would increase the chances of discovering interesting patterns or relations within these complex fields. The dashboards of this case study have a series of particular requirements due to the data domain and the specific characteristics of the Observatory studies. For instance, the developed data visualizations exploit different

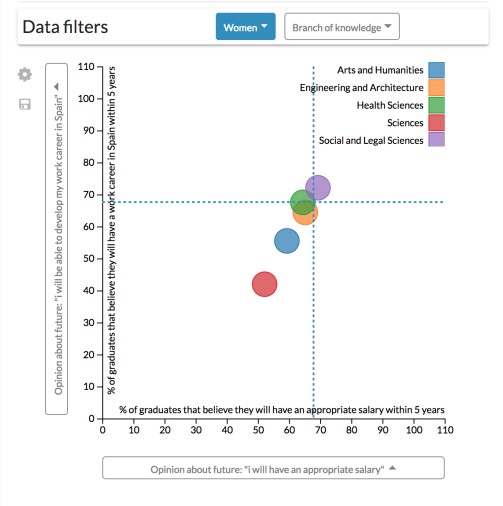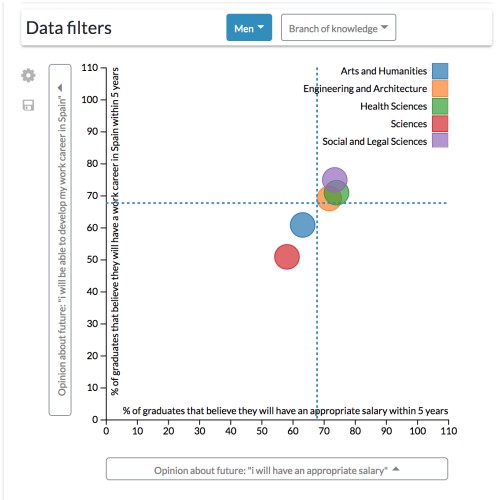

**Figure 11** **Results derived from the first configuration.** Through this configuration is possible to apply different filters simultaneously to each scatter diagrams to observe how patterns evolve.

dimensions of the Observatory's collected variables. Also, the generated Observatory's dashboards needed to be connected to the organization's GraphQL API (*Facebook, 2016*) that allow users to retrieve data statistics on demand (*Vázquez-Ingelmo, Cruz-Benito & García-Peñalvo, 2017*; *Vázquez-Ingelmo, García-Peñalvo & Therón, 2018a*; *Vázquez-Ingelmo, García-Peñalvo & Therón, 2018b*; *Vázquez-Ingelmo, García-Peñalvo & Therón, 2018c*), decoupling the data resources from the visual components' logic.

In this section, the results derived from the application of the presented dashboard product line within the university employment and employability domain are described. By tuning SPL through particular configurations, it is possible to obtain tailored solutions for different requirements and tasks.

**Configuration #1.** Comparison of different values is one of the most relevant tasks regarding the exploration of university employability and employment data. These comparisons could enlighten which factors affect employability and employment to a greater or lesser extent, leading to the possibility of conducting deeper analyses.

For example, by configuring a dashboard with two scatter diagrams side by side, it is possible to apply different filters to each one and observe how data patterns evolve (Fig. 11). Also, adding the global reference feature to both diagrams helps to make comparisons by adding a reference line marking the unfiltered and disaggregated values.

It is possible to appreciate, for example, that men graduates are more optimistic when commenting opinions about their future wages and the possibility of developing a working career in Spain (*Michavila et al., 2018b*). However, these diagrams also allow seeing at a glance that Arts and Humanities and Sciences graduates are more pessimistic about their future than their counterparts in other branches of knowledge, which are more clustered. For instance, only 40% of Sciences women graduates think that they could have a working career in Spain within five years.

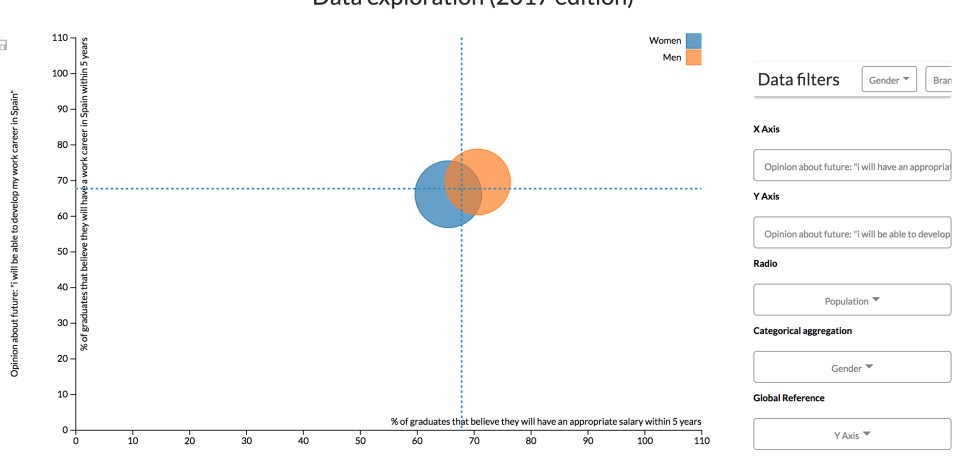

**Figure 12** **Results derived from the second configuration.** The scatter diagram shows the link between different students' opinions classified by gender.

This configuration enables the user to explore data through the combination of different aggregations, variables and filters.

**Configuration #2.** The previous configuration, however, could be complex for some users by having to control two diagrams at the same time to align different factors. A single scatter diagram could be added to the dashboard to drill-down data. It is possible to add another dimension to the scatter diagram component by mapping numerical variables through the radius of the visualization's data points.

For instance, following the same example of the first configuration, the differences between male and females can be observed by a gender aggregation of the data. In this case, the population of each group is mapped through the radius of the points (Fig. 12).

However, to see how the branch of knowledge affects the value of these variables, similarly to the previous configuration, it is necessary to continuously filter data by every single branch (Fig. 13). This configuration is then not recommendable when continuous, and more complex comparisons (such as the one made in the previous scenario) are required.

If, on the other hand, data exploration is not continuously required by a user, the controls could be allocated within a top bar (Fig. 14) that can be hidden to give more space to the visualizations.

**Configuration #3.** On the other hand, different pages focused on different data variables or data dimensions could be configured. This functionality allows freedom when arranging the content of the dashboards' pages to make it understandable for every particular user.

In the Observatory's case, a user might prefer having the dashboard screens organized by the study edition, being able to navigate through them thanks to a navigation bar (Fig. 15).

Or if preferred, it could be specified that each page will exploit a different set of data variables; for example, having a single tab to explore the students' competences (Fig. 16).

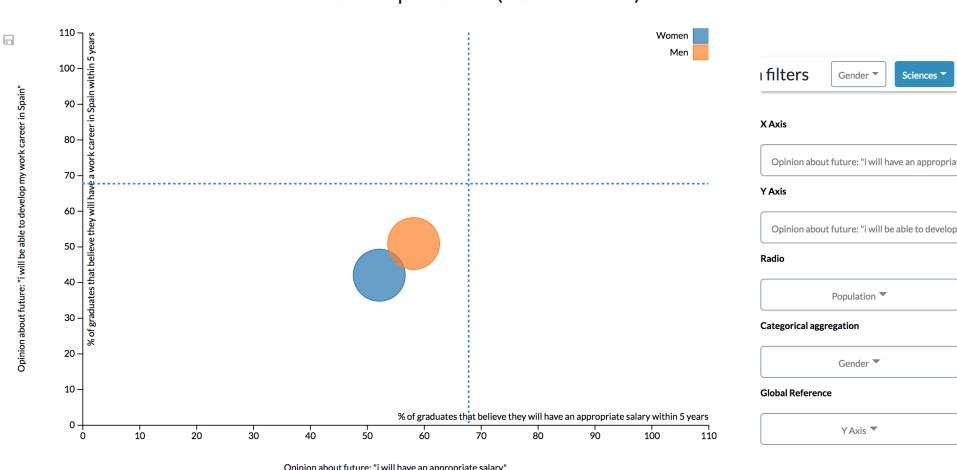

**Figure 13** **Results derived from the second configuration.** The scatter diagram shows the link between different students' opinions classified by gender and filtered by the branch of knowledge, showing only the results related to Science students.

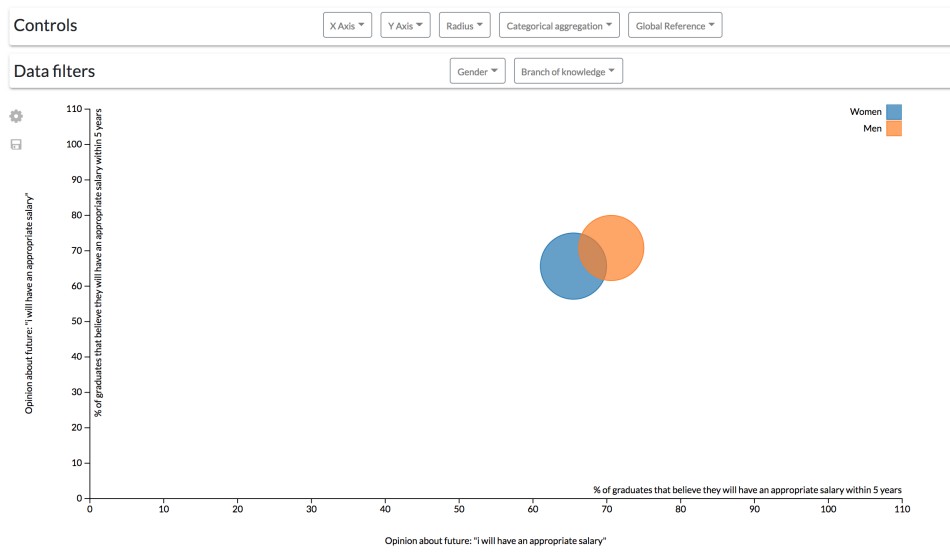

**Figure 14** **Modification of the second configuration to change the controls location.** The controls for the scatter diagram are arranged in a bar on top of the visualization.

Through this view it is possible to see a misalignment between the perceived level that the graduates have about their skills and the perceived level of contribution of the studies regarding the acquisition of that skills, and also between that possessed level and the perceived required level in their job positions (*Michavila et al., 2018b*).

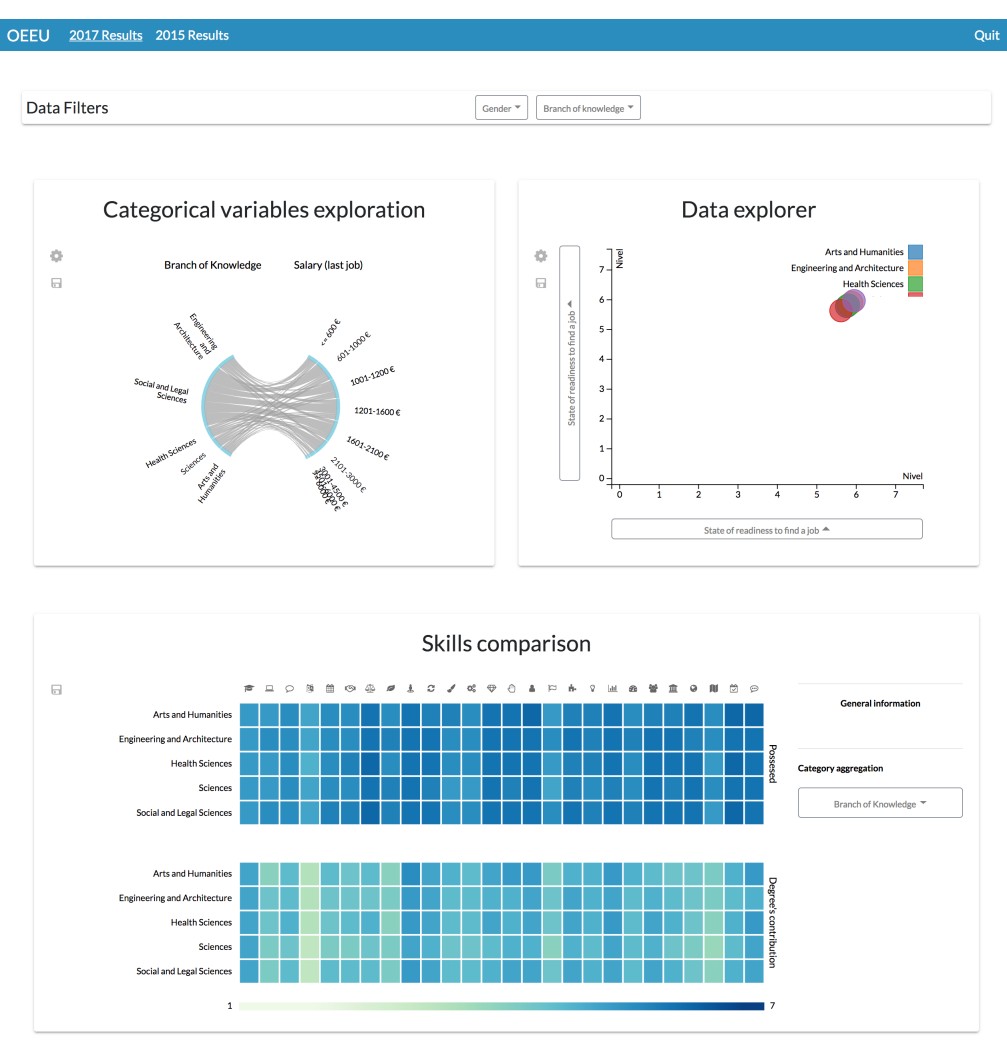

**Figure 15** **Dashboard involving different information visualizations.** By specifying the layout of the dashboard it is possible to achieve dashboards with different components, each one with its own features.

The previous dashboards are a quite tiny set of the available combinations that can be achieved through the SPL configuration, but they should serve as an example to show the possibilities of having a framework for generating personalized dashboards.

## Product metrics

The metrics for the SPL are the following regarding its feature model:

- Feature model height: 9
- Features: 146
- Optional features: 106

The number of valid configurations has been omitted, given the recursion of the dashboards' composition (as highlighted in the dashboard meta-model), so infinite valid configurations can be generated.

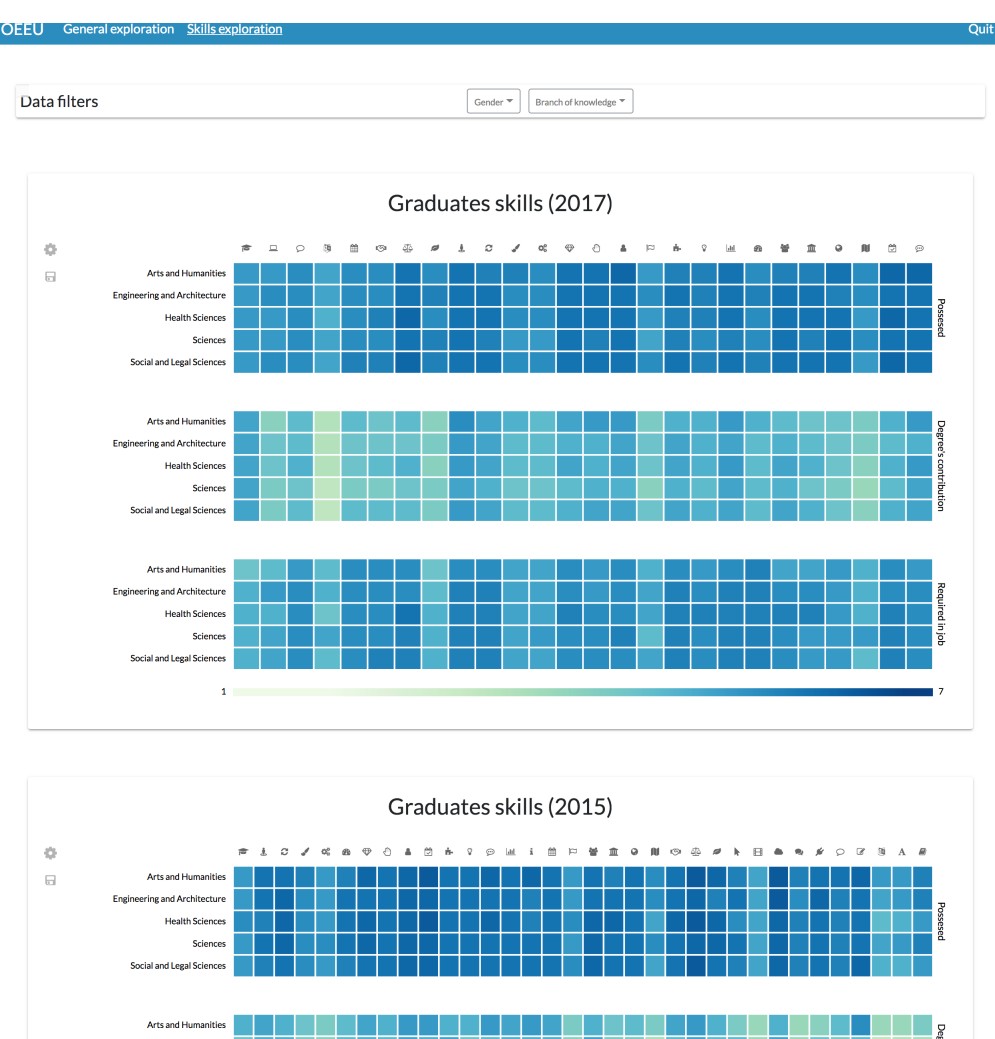

**Figure 16   Possible layout configuration for comparing students' skills through different study editions.** This configuration can be useful to identify lack of skills at-a-glance or their evolution through time.

Regarding the core-assets (i.e., the templates' source code), the following metrics have been calculated (*El-Sharkawy, Yamagishi-Eichler & Schmid, 2018*):

• Lines of feature code (LoF): 2,638 lines of feature code. This metric is the addition of every line of code affected by any Jinja2 directive (i.e., every annotated line of code). It is a size metric that gives a high-level view about the source code associated to the SPL features.

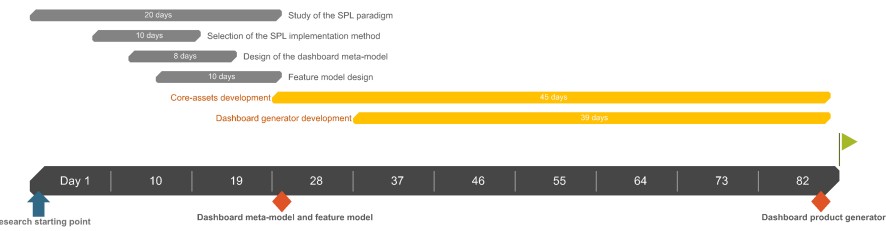

**Figure 17** **Simplified Gantt diagram of the SPL development.** The Gantt diagram shows each task regarding the SPL development including its contextualization and design.

• Fraction of annotated lines of code (PLoF): 48.39%. This is a variability density metric showing that the SPL's products have a 51.61% of common code (2,814 lines of code are not annotated).

• Scattering of variation points: this metric counts the number of times that a feature appears in the code (i.e., appears in a Jinja2 condition directive). High scattering values decreases the readability of the code. By refactoring the code into macros that contain all code associated to a specific feature, the scattering is reduced.

Given the complex domain in which the product line has been applied (i.e., the dashboards' domain), the scattering of the variation points was one of the main concerns, as high scattering would make the code even more complex. That was the reason to arrange the feature code into macros as a solution to address the scattering of variability points.

## Development time improvement

The development of the presented SPL, including its conceptualization and design, took 82 days, as illustrated through a simplified Gantt diagram in Fig. 17. The core assets development task includes all the artifacts regarding the SPL (i.e., the DSL, the templates, etc.).

Before implementing this approach, a dashboard template with the same components and KPIs was the solution to offer all the results held in the Observatory's study, so universities could compare their individual results with the global, aggregated results. The development of the mentioned dashboard template took 15 days. However, this static approach limited universities to freely explore their data, as mentioned in other sections.

Five of the 50 universities were interviewed to capture their dashboard requirements and to estimate the elicitation process time consumption. However, this estimation should be considered as speculative given the variability of the complexity of the elicitation process, and especially, given the number of different universities (i.e., users) involved. Nevertheless, the requirement elicitation took one day for the interviewed universities.

Given the project's potential continuity, the dashboard implementation process would mainly consume time regarding requirements elicitation by using the presented SPL approach, decreasing the time spent on development processes. Without this approach, the information dashboards implemented for future Observatory's employability study editions would remain static and generalized for each involved user.

Building a personalized dashboard consume resources in terms of requirement elicitation and design, but also in terms of implementation or development. If the development phases are automated, then the main benefit is not only decreasing the development time of individual dashboards, but also, if necessary, devoting more time to the requirements identification and design phases, which, in the end, are the backbone of well-constructed dashboards. That is why, although significant time was consumed for the implementation of the dashboard SPL (82 days), it can be seen as an investment for the future, specifically in environments where significant quantities of user profiles are involved.

## DISCUSSION

The application of domain engineering and the SPL paradigm to identify and factorize information dashboard functionalities has shown its usefulness to generate different dashboards with a set of common assets through the study of the dashboards' domain. The obtained results are fairly valuable, and open new paths for applying this approach to other data domains with new requirements.

Dashboards are complex software solutions that could be highly beneficial when adequately designed and tailored for specific users. These products can support decision-making processes, assisting visual analysis by presenting information in an understandable manner. However, the variety of profiles involved in these processes and their different definitions of "understandable" makes the implementation of dashboards a time- and resource-consuming task, since a dashboard configuration that is highly useful for one user could be pointless for the rest of them. What is more, dashboards can be composed of several elements, from simple visualizations to different linked views, cross-filtering capabilities, interaction methods, handlers, etc., thus making the dashboards' domain a complex domain not only because of the different profiles of potential users, but because of the great quantity of feasible combinations of these "dashboard elements" to build a proper solution. In addition, these features can be very fine-grained; in user-centered systems, a slight modification on visualization types, interaction patterns, layouts, color palettes, etc. could be crucial regarding the final perceived usability of the product.

Relying on a framework to easily generate information dashboards would allow stakeholders to focus on the information requirements and their refinement to provide better results when seeking valuable insights on large datasets. Also, it opens up the possibility to automatically adapt the dashboards' configurations to match dynamic requirements based on the device used (*Cruz-Benito et al., 2018b*) or other factors.

The factorization of the dashboards' components into individual features allow fine-grained reusability and a set of customization options. This fine-grained customization enables the possibility of having highly functional and exploratory-centered visualizations as well as more basic visual components more centered on the explanation of insights through the addition or removal of low-level features. The achieved granularity provides a foundation to develop not only whole visualization components, but also new interaction methods and design features that can be easily interchangeable to fulfill particular sets of user requirements.

An annotative method of implementation was undertaken using macros to encapsulate individual functionalities. This method takes all the benefits from the annotative approach (fine-grained customization) and avoids its code verbosity and scalability issues by dividing the core assets into base templates and macros (*Kästner & Apel, 2008*). Although there were other possibilities to implement the variability points, such as superimposition approach (which did not fulfilled the requirements for performing this approach, as discussed in the Materials & Methods section) like the FeatureHouse framework (*Apel, Kastner & Lengauer, 2009*; *Apel & Lengauer, 2008*) or the XVCL (*Jarzabek et al., 2003*) mechanism (which fits the feature granularity requirements of this domain), the final decision of using a templating engine allowed the direct connection of the designed DSL with the final source code, providing a higher level language to specify the dashboards' features, as well as the possibility of organizing the variability points into macros to increase readability, traceability and maintainability by having all the code associated to a feature in the same source file.

The chosen technology to implement the DSL was XML. The decision of implementing directly the DSL in XML technology was made because of the hierarchical nature of XML, and its resemblance to the hierarchical structure of the feature diagram, thus being the designed DSL a computer-understandable "translation" of the feature model for the dashboard generator to process. However, this language could be not as human-readable as other DSL solutions, generating issues if a non-expert user wants to specify its dashboards requirements by himself. Creating a friendly user interface to allow the dashboards' feature selection without involving direct manipulation of the XML files can be a valuable solution to address these issues and ease the product configuration process in the future.

Customization at functionality level has illustrated to be straightforward, as it is possible to easily vary the behavior of the visual components through the DSL. Visual design attributes customization, however, needs to be faced more deeply, as only the layout composition can be specified in detail at the moment. The visual customization challenge cannot be overlooked since dashboards not only have to provide valuable functionality; they should offer that functionality through a pleasant and usable interface (*Few, 2006*; *Sarikaya et al., 2018*).

On the other hand, this work has addressed customization focused on the presentation layer of dashboards, but with the SPL paradigm, architectural design can also be customized in order to provide different functional features regarding data processing, interoperability, storage, performance, security, etc., achieving a complete customizable dashboard solution, not only focusing on the visual presentation.

Regarding data acquisition, the developed tool was integrated with the Observatory's GraphQL API to provide dynamic data exploration. The connection to this particular type of data source involved the implementation of specific connectors to decouple the visualizations from the particular source. The variability of data sources is another identified challenge to be addressed through this approach, to support different data formats or data structures. Although counting on a GraphQL API facilitated the data retrieval by the unification of data requests, it is essential to enable the specification of other data retrieval methods.

Product metrics showed that significant feature code was needed to address high customizability of the dashboards (48.39% of the source code was annotated). Also, arranging the feature code into macros helped to increase features' traceability as well as to decrease the scattering of the variability points throughout the code, making the code more readable and maintainable.

The approach can decrease the development time of individualized dashboards for each involved university. As presented in the results section, the SPL not only offered space for development time improvements, but also enabled the capacity of offering customized solutions, which was previously regarded as unviable given the time constraints of the Observatory's project. Embracing the SPL paradigm can be seen as an investment for the future for projects with a common domain and with continuity over time.

Finally, it is clear that interesting patterns can be discovered thanks to the application of this dashboard SPL on the employability and employment fields. The Observatory's data provide a great context to perform more advanced analyses to enlighten this complex domain.

Having powerful visualization tools allow reaching insights about patterns or factors to guide the execution of more complex analyses and make decisions about the actions to take or the future research directions, like developing machine learning (ML) models (*García-Peñalvo et al., 2018*). Regarding this last field, having visualization tools to explore the input data before training any ML model could help to build better and more accurate models through an appropriate feature selection phase guided by the previously reached insights (*Hall, 1999*).

The main weaknesses and limitations of this solution come from the preliminary nature of the framework; it is crucial to further validate the usability of the automatically generated products to show their usefulness to the main beneficiaries of the dashboards: the users, as well as assess its implementation in other domains. The approach needs to be further generalized to provide a more versatile method and to match also development requirements (available technology or preferred programming languages), although results seem promising. Automating the generation of dashboards given their goal, their context, their end users, etc. could be extremely beneficial due to the vast potential of impact that these tools have (*Sarikaya et al., 2018*).

## CONCLUSIONS

A domain engineering approach has been applied to the dashboards' domain to obtain a SPL of this type of software solution. By the identification of commonalities and variability points, a dashboard meta-model has been developed as well as a feature model to capture the different customization dimensions of the SPL.

The SPL has been developed through an annotative approach using code templates and macros (forming the core assets of the family of products). A DSL has been designed to facilitate and automate the application engineering process. The configuration files based on the DSL feed a code generator in charge of adding or removing the product features. The presented approach was applied within the Spanish Observatory for

University Employability and Employment system, to provide a variety of dashboard configurations that enable the exploitation and exploration of different dimensions regarding employability and employment data.

Future research lines will involve refinements of the meta-model and the DSL, usability testing of the obtained products and A/B testing (*Cruz-Benito et al., 2018a*; *Cruz-Benito et al., 2018b*; *Kakas, 2017*; *Siroker & Koomen, 2013*) on different configurations. Architectural customization could be supported to add more coarse-grained features like a visualization recommendation engine (*Gotz & Wen, 2009*; *Vartak et al., 2017*; *Voigt et al., 2012*), interface language translation or data preprocessing techniques before its presentation. Finally, the customization levels of the dashboards' visual design and data sources need to be further addressed.

### Funding

This work was supported in part by the Spanish Government Ministry of Economy and Competitiveness throughout the DEFINES project (Ref. TIN2016-80172-R), in part by the PROVIDEDH project, funded within the CHIST-ERA Programme under the national grant agreement: PCIN-2017-064 (MINECO, Spain) and in part by La Caixa Foundation. The work of A Vázquez-Ingelmo was supported by the Spanish Ministry of Education and Vocational Training under an FPU fellowship (FPU17/03276). There was no additional external funding received for this study. The funders helped with the contextualization of the research.

### Grant Disclosures

The following grant information was disclosed by the authors:
Spanish Government Ministry of Economy and Competitiveness: Ref. TIN2016-80172-R.
CHIST-ERA Programme: PCIN-2017-064.
La Caixa Foundation.
Spanish Ministry of Education and Vocational Training: FPU17/03276.

### Competing Interests

The authors declare there are no competing interests.

### Author Contributions

- Andrea Vázquez-Ingelmo conceived and designed the experiments, performed the experiments, analyzed the data, contributed reagents/materials/analysis tools, prepared figures and/or tables, performed the computation work, authored or reviewed drafts of the paper.
- Francisco J. García-Peñalvo and Roberto Therón conceived and designed the experiments, performed the experiments, analyzed the data, contributed reagents/-materials/analysis tools, performed the computation work, authored or reviewed drafts of the paper, approved the final draft.

## Data Availability

Data and assets are available at https://github.com/AndVazquez/dashboard-spl-assets. (DOI: 10.5281/zenodo.1478134).

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
