# Peer review of "Taking advantage of the software product line paradigm to generate customized user interfaces for decision-making processes: a case study on university employability"

_PeerJ Computer Science, doi:10.7717/peerj-cs.203_

## Round 0.1 · original submission · Major Revisions

The two reviewers have provided very detailed and helpful feedback on your current submission. Please follow their advice carefully - in particular you need to define your research problem more clearly and present the evidence to answer it more clearly. I hope the reviewers will be useful to you in moving your research forward.

Reviewer 1 ·

Basic reporting

The article is generally well written, with good spelling and grammar. There examples of terms that I myself would probably not use e.g. l388 "can be also consulted at". I would just word it as "can be viewed at".

There are a few works that I believe are very relevant that were missing. These include:

1) Apel, S., Kastner, C. and Lengauer, C. (2009). Featurehouse: Language-independent,
automated software composition, Proceedings of the 31st International Conference on Software Engineering, ICSE ’09, IEEE Computer Society, Washington,
DC, USA, pp. 221–231.

2) Kramer D., Oussena, S., Komisarczuk, P., Clark, T. (2013) Document-Oriented
GUIs in Dynamic Software Product Lines. In the Proceedings of the 12th International Conference on Generative Programming: Concepts and Experiences.

3) Kramer D., Oussena S. (2017) User Interfaces and Dynamic Software Product Lines. In: Sottet JS., García Frey A., Vanderdonckt J. (eds) Human Centered Software Product Lines. Human–Computer Interaction Series. Springer

4) Pleuss, A., Hauptmann, B., Dhungana, D. and Botterweck, G. (2012). User interface engineering for software product lines: the dilemma between automation
and usability, Proceedings of the 4th ACM SIGCHI symposium on Engineering
interactive computing systems, EICS ’12, ACM, New York, NY, USA, pp. 25–34.

5) Pleuss, A., Hauptmann, B., Keunecke, M. and Botterweck, G. (2012). A case study
on variability in user interfaces, Proceedings of the 16th International Software
Product Line Conference - Volume 1, SPLC ’12, ACM, New York, NY, USA, pp. 6–
10.

Structure of article is generally fine. I am unsure why code samples are given as images. Data shared in github repository. Results seem speculative. There seems to be no test or analysis if a SPL approach is actually beneficial to the papers context.

Experimental design

The context in which the experiment is being undertaken is original, as is the approach. As there are a number of works missing (as mentioned above), it is difficult to follow why the SPL approach was applied. There are opensource tools e.g. FeatureHouse that allow for fine grained software composition. The generator created for this experiment appears to be specific for this SPL, and can not be used for another SPL. Tools like FeatureHouse are generic, and can be applied to many different types of SPLs, with many different languages. There is no discussion of this. Many of related work included reads like an acknowledgement, however a lack of real motivation as to why the GUI approaches used are not sufficient to their use case.

The research question for me is not well defined. On the one hand, they argue for the need for customisation to support different user needs, but also argue on the need for improving productivity, maintainability, and traceability. The needs for supporting customisation in this context is interesting, however I see no evaluation of this. Were the different product variants of use to the users? Did the students find they picked the right courses for them?
It does not feel like that research questions were answered. There is some discussion on some sample variants produced by the SPL, but no discussion on the real benefits of the SPL in the first place.

Validity of the findings

The results given are regarding specific variants produced from the SPL, without answering the questions motivated by this paper. Findings appear to be qualitative by the authors, not from users, developers (that many use the SPL), or quantitative following existing approaches.

It is state on line 476-478 argue the SPL has "proved its usefulness to reduce the development time regarding the exploration of the observatory's data". How was this tested? Where are the results to support that claim?

Additional comments

You need to really explain more concretely your research problem, and then conduct experimentation to help answer those questions. I honestly struggle to see how the results you show answer the questions I have interpreted from your paper. How has using an SPL really helped you? Why is your approach better/novel to related works? How did the users evaluate your variants? You results only seem to be talking about some of the configurations, but appear to not help answer your research questions.

Reviewer 2 ·

Basic reporting

The background section contains a good range of related work, but there is no proper comparison with the work proposed. The related works are reported, without any relation to the proposal. This would benefit the text, in making clear what are the main contributions of this work compared to previous ones, where are previous works falling short. It is not clear if dashboards are configured statically, that is, in a compile-time fashion. Couldn’t works related to dynamic software product lines be useful on this context, to provide dynamic configuration and customisation for the dashboards?

- When discussing explanatory and exploratory visualisations (lines 235-243), it would be good to provide concrete examples for illustration.

- When discussing the meta-model (line 286), this is general to any dashboard, not only for those related to employability or those who are customisable, correct?

- Repeated use of “proved” in the text, when it should be used ‘illustrated’, ‘demonstrated’, ‘shown’, or some other term.

- In the beginning of the context section, line 205: “…also known as OEEU, it’s Spanish…” -> “…also known as OEEU, its Spanish…”

- What does “complete and quality information” mean in the introduction? (line 78)

- Line 230: Does the text actually mean ‘recollected’ or ‘collected’ would suffice?

- Line 272: …abstract point of view… —> abstract modelling?

- Line 400: ‘sets of data’ -> ‘datasets’

- Some of the references are incomplete, without edition and pages for journal papers, for instance. Moreover, there are works mentioned in the text, such as Gabillon et al. (2015) and Pleuss et al. (2012), which are not listed on the references section

Experimental design

Nonetheless, the text in its current version does not clearly present the research question. The motivation properly addresses the issue of a range of definitions for employability, with different perspectives. Nonetheless, something that is not clear in the current version of the text is the fact that customisable dashboards could make that even more confusing and diverse. Wouldn’t that make it hard to have a common, shared definition for the concept?

I understand that for space constraints, the feature model is not provided, but I believe that at least the repository should have a version of it available. Another thing that would benefit and help on motivating the tool support developed on this work, would be to calculate the actual number of products that can be generated from the selection of features from the model. This could be done with the aid of tools such as FAMA, FAMILIAR, or FeatureIDE, and would help on establishing the need for having a systematic and automated process and avoiding manual labour into customising the dashboards.

Another issue is related to the particular notation for feature models that was used. On Figure 3, for instance, there is a feature named Size? Would that entail an attribute for the size, or does it make sense as a simple boolean feature that might be optionally present or not? Another question would be the ‘categories’ feature? Would it be necessary to list the categories for a specific dataset, or does it suffice to enable/disable it as a whole?

It seems to me that the DSL shown in the work is an embedding of the feature model in XML. If that’s the case, this should be made explicit in the text. Also, why not using a proper DSL that could then generate the underlying XML files as a result? In terms of traceability of the DSL with the feature model, how is that maintained? In the github repository provided as raw data, there is only the schema, and no file, even in image form, depicting the feature model. In the end, do we actually need the feature model in this approach? Or are DSL files being automatically generated from the feature model?

Figure 9 shows an instance of the code generation process, it would be better to first present something similar as Figure 10, but with proper identified roles (see my other comment below on who’s responsible for performing the customisation), and then illustrating it with the concrete example from Fig. 9. I also believe that less detail could be provided, so text could be made bigger and more readable.

Another issue is related to the actual process of configuration and customisation. Who is in charge of configuring the dashboard, the actual user or someone prior to the end-user of the system?

Did the work need to perform any changes to Jinja, or could it be used as-is?

Validity of the findings

The discussion section starts by stating that the SPL paradigm has ‘proved’ (see comment about this term) the usefulness to reduce the development time. While an automated process can surely help on reducing the time for development, currently there is no evidence presented in the text to support this. What was the time needed for developing the DSL and whole infrastructure for generating products? Usually, in the literature, there is a return on investment for the product line engineering approach starting from three (3) products. Was that observed from the case study?

I believe that the discussion section and results should focus on effort (even if estimated) related to this approach vs. a manual one. Even proxy metrics, such as lines of code could be used for comparison. Another issue would be to try considering evaluating expressiveness? Are there any limitations on the current approach which prevent a particular customisation to be performed? The text mentions this by the end of the discussion section, but I believe that an immediate follow-up of this work should focus on end-users evaluating the usability of the generated dashboards to confirm whether those are actually usable for analysing the data.

Additional comments

This article illustrates the application of a software product line engineering approach to the development of information dashboards geared towards analysing data regarding university employability. Through the use of XML-based DSLs, a range of customisable dashboards can be automatically generated. The text provides some examples of custom configurations and discusses the main issues related to applying the SPL approach.

The work brings the application of software product lines to the domain of generating information dashboards, with a focus on customisable user interfaces, for exploring large datasets. However, there are a number of issues in the current version of the text, which I detail in the three (3) dedicated areas.

---

## Round 0.2 · Minor Revisions

Thank you for addressing the major changes suggested by the reviewers. We have considered the revised manuscript and are of the opinion that you have done a good job of addressing most of the points made. However, the paper would be greatly improved by a small number of minor improvements. These are detailed below.

Reviewer 2 ·

Basic reporting

I appreciate the changes performed to the manuscript as requested by the reviewers.

- Line 119: Viability - feasibility

- Line 643: “to prove…” - “to show…”

- Line 644: “The approach need…” - “The approach needs…”

Experimental design

Even though there have been further explanations, I would still suggest better discussing the integration of feature diagrams and the DSL, besides justifying why not using a proper DSL that could then generate the underlying XML files as a result.

Regardless of what is currently implemented in the approach, does it conceptually need a feature model, or is it just a convenience for having a readable document? That’s related to what I had asked in the prior revision, so I would expect something along these lines in the text.

Validity of the findings

The “development time improvement” section is a good addition. I would only suggest putting a stronger emphasis in the fact that even though there is effort to setup the SPL, this might be paid off by easily generating customisable dashboards.

In the final discussion, I would also suggest highlighting what is unique in the domain of complex tools, such as dashboards, that has been tackled by the approach.

Additional comments

This article illustrates the application of a software product line engineering approach to the development of information dashboards geared towards analysing data regarding university employability. Through the use of XML-based DSLs, a range of customisable dashboards can be automatically generated. The text provides some examples of custom configurations and discusses the main issues related to applying the SPL approach.

---

## Round 0.3 · accepted · Accept

Thanks for persisting with the requests for improvements. I think the paper is now much better.